# Value of D-dimer in predicting various clinical outcomes following community-acquired pneumonia: A network meta-analysis

Jiawen Li[ID][☯], Kaiyu Zhou[☯], Hongyu Duan[☯], Peng Yue, Xiaolan Zheng, Lei Liu, Hongyu Liao, Jinlin Wu, Jinhui Li, Yimin Hua*, Yifei Li[ID]*

Department of Pediatrics, Key Laboratory of Birth Defects and Related Diseases of Women and Children of MOE, West China Second University Hospital, Sichuan University, Chengdu, Sichuan, China

☯ These authors contributed equally to this work.
* Nathan_hua@163.com (YH); liyfwcsh@scu.edu.cn (YL)

**Data Availability Statement:** All relevant data are within the manuscript and its Supporting Information files.

## Abstract

### Background

Whether high D-dimer level before treatment has any impact on poor outcomes in patients with community-associated pneumonia (CAP) remains unclear. Therefore, we conducted the first meta-analysis focusing specifically on prognostic value of high D-dimer level before treatment in CAP patients.

### Methods

Pubmed, Embase, the Cochrane Central Register of Controlled Trials and World Health Organization clinical trials registry center were searched up to the end of March 2021. Randomized clinical trials (RCT) and observational studies were included to demonstrate the association between the level of D-dimer and clinical outcomes. Data were extracted using an adaptation of the Checklist for Critical Appraisal and Data Extraction for Systematic Reviews of Prediction Modeling Studies (CHARMS-PF). When feasible, meta-analysis using random-effects models was performed. Risk of bias and level of evidence were assessed with the Quality in Prognosis Studies tool and an adaptation of Grading of Recommendations Assessment, Development, and Evaluation. Data were analyzed using STATA 14.0 to complete meta and network analysis.

### Main outcomes and measures

Besides d-dimer levels in CAP patients with poor outcomes, we also analyzed proportion of patients with or without poor outcomes correctly classified by the d-dimer levels as being at high or low risk. The poor outcome includes severe CAP, death, pulmonary embolism (PE) and invasive mechanical ventilators.

### Results

32 studies with a total of 9,593 patients were eventually included. Pooled effect size (ES) suggested that d-dimer level was significantly higher in severe CAP patients than non-

**Funding:** This work was supported by grants from the National Natural Science Foundation of China awarded to YL (81700360) and HD (81971457, 81602817). The funding source had no role in the design and conduct of the study; collection, management, analysis, and interpretation of the data; preparation, review, or approval of the manuscript; and decision to submit the manuscript for publication.

**Competing interests:** The authors have declared that no competing interests exist.

**Abbreviations:** CAP, community-acquired pneumonia; ELISA, enzyme-linked immunosorbent assay; ICU, intensive care unit; PE, pulmonary embolism; PSI, pneumonia severity index; QUIPS, quality in prognostic studies; SUCRA, surface under the cumulative ranking; SMD, standard mean difference; SD, standard deviation; HSROC, hierarchical summary receiver operating characteristic curve.

severe CAP patients with great heterogeneity (SMD = 1.21 95%CI 0.87–1.56, $I^2$ = 86.8% p = 0.000). D-dimer level was significantly elevated in non-survivors compared to survivors with CAP (SMD = 1.22 95%CI 0.67–1.77, $I^2$ = 85.1% p = 0.000). Prognostic value of d-dimer for pulmonary embolism (PE) was proved by hierarchical summary receiver operating characteristic curve (HSROC) with good summary sensitivity (0.74, 95%CI, 0.50–0.89) and summary specificity (0.82, 95%CI, 0.41–0.97). Network meta-analysis suggested that there was a significant elevation of d-dimer levels in CAP patients with poor outcome than general CAP patients but d-dimer levels weren't significantly different among poor outcomes.

## Conclusion

The prognostic ability of d-dimer among patients with CAP appeared to be good at correctly identifying high-risk populations of poor outcomes, suggesting potential for clinical utility in patients with CAP.

## Introduction

Community-acquired pneumonia (CAP) is defined as pneumonia acquired outside the hospital and has led to life-years lost globally [1]. Approximately 6.6% to 16.7% of hospitalized patients with CAP would enter the severe stage. Unfortunately, mortality rate is supposed to reach up to 28.8% among patients with severe CAP [2]. A considerable proportion of patients with CAP in the emergency department can be treated as outpatients. However, unpredictable disease course and uncertain outcomes are challenges for clinicians, hindering the early identification of patients at high risk. Several risk scores, such as the pneumonia severity index (PSI) and CURB-65 (confusion, urea nitrogen, respiratory rate, blood pressure, age ≥65 years), can be used to assess the severity of pneumonia and predict prognosis [3–5]. For CAP, low risk was defined as PSI score classes I to III and CURB-65 score class 1. High risk was defined by PSI score classes IV-V and CURB-65 score classes 2–5. However, they were more suitable for research than clinical decision and their performance is still controversial [6]. C-reactive protein (CRP) [7, 8] and procalcitonin (PCT) [9, 10] had been reported be a prognostic marker of outcome during severe CAP and ventilator-associated pneumonia (VAP). D-dimer is the fibrinolytic degradation products of crosslinked fibrin and is applied as a useful marker for the diagnosis of pulmonary embolism. It mirrors the severity of infection and has emerged as the extensively studied and promising blood biomarker for the risk stratification of patients with CAP [11–13].

However, whether d-dimer level is an ideal index to predict the prognosis of community-acquired pneumonia remains debatable [14, 15], because no relevant studies focusing on d-dimer levels before treatment in patients with CAP were available to conduct meta-analyses previously. For this reason, a meta-analysis was performed to systematically and quantitatively evaluate the prognostic accuracy of the d-dimer level before treatment in CAP. To our knowledge, this is the first meta-analysis specifically focusing on d-dimer levels in patients with CAP. In consideration of different disease process and different therapeutical strategies of COVID-19 due to its distinct biology and pathogen, we haven't included publications of COVID-19 to prevent unsolvable heterogeneity.

## Materials and methods

### Study protocol

This analysis was conducted in accordance with a predetermined protocol following the recommendations of a guideline for systematic reviews of prognostic factor studies [16]. And the data collection and reporting was in accordance with Preferred Reporting Items for Systematic Reviews and Meta-Analyses: The PRISMA Statement [17] and Extension Statement for Reporting Network Meta-Analyses [18]. (PROSPERO; CRD42020184704)

### Search strategy

We searched the Pubmed, Embase, the Cochrane Central Register of Controlled Trials and World Health Organization clinical trials registry center using a comprehensive strategy to get the publications. The strategy was "(("pneumonia"[MeSH Terms] OR (community-acquired [All Fields] AND "pneumonia"[MeSH Terms])) OR ("pneumonia"[MeSH Terms] OR "pneumonia"[All Fields])) AND ("fibrin fragment D"[Supplementary Concept] OR "fibrin fragment D"[All Fields] OR "d dimer"[All Fields])". Search was updated to the end of March 23, 2021 with language restricted to English.

### Study selection

Titles and abstracts of search results were screened independently (Jiawen Li, Hongyu Duan). The full texts of the remaining results were assessed independently by another 2 of us (Yifei Li, Yimin Hua) for inclusion based on predetermined criteria. Any discrepancies should be resolved through discussion, potentially with a third reviewer. We manually searched the reference lists of included studies and existing systematic reviews as well as all articles citing the included studies on Google Scholar. Potentially relevant reports were then retrieved as complete manuscripts and assessed for compliance to inclusion and exclusion criteria.

In accordance to the objectives of our meta-analysis, we developed a 'Population, Index prognostic factor, Comparator prognostic factor, Outcome, Timing, Settings' (PICOTS) framework adapted from the guideline proposed by Riley et al [16]. Our study inclusion criteria were as follows according to PICOTS framework: (1)Population: CAP patients with a well-defined diagnostic reference standard for pneumonia; (2)Index prognostic factor: before-treatment d-dimer levels measured by enzyme-linked immunosorbent assay (ELISA), quantitative latex assay, immunoturbidimetric or other convinced assay machine; (3)Outcome: severe CAP patients identified by PSI, CURB-65 or other definitive scale, pulmonary embolism (PE), death or invasive ventilation. (6) If studies were based on overlapping patients, the most completed one was chosen. We used the following criteria for study exclusion: (1) patients acquired pneumonia in clinical settings (e.g. VAP or hospital-acquired pneumonia (HAP)); (2) studies were published in other language; (3) conference abstracts, reviews, case reports, and experiment studies.

### Data extraction and study quality assessment

Two reviewers (Jiawen Li, Hongyu Duan) independently extracted study data and assessed risk of bias, with discrepancies resolved by a third investigator in a blinded fashion. Quality of evidence was assessed by the modified Grading of Recommendations Assessment, Development, and Evaluation system (GRADE) by consensus among the authors [19, 20].

The essential data was extracted according to the modification of CHARMS (checklist for critical appraisal and data extraction for systematic reviews of prediction modelling studies) for prognostic factors (CHARMS-PF) [16]. When an included study reported different cut-off

values, we chose one which made both sensitivity and specificity more than 50% as possible. When an included literature reported the same outcome at different follow-up timepoint (e.g. 7-days mortality and 30-days mortality), we chose the earliest one. We extracted data of PSI if included studies reported severity of CAP by both PSI and CURB-65. Mean and standard deviation were estimated based on sample size, median and quartile if included studies did not reported mean and standard deviation [21, 22]. All the baseline characteristics of included studies were shown on Table 1.

The included studies were further investigated for risk of bias using an adapted version of the Quality in Prognosis Studies (QUIPS) tool [23], which assessed the study-specific risk of bias across to six bias domains: study participation, study attrition, prognostic factor measurement, outcome, measurement, study confounding, statistical analysis and reporting. A study that satisfied the criteria of low risk of bias in all 6 domains was designated as having low overall risk of bias. A study with a high risk of bias in 1 or more domains was designated as having high overall risk of bias. Details on each signaling question of the QUIPS tool are elaborated on Table 2. We did not exclude any publication with high risk of bias according to QUIPS.

## Statistical analysis

Analyses were performed for both adjusted and unadjusted estimates. To combine comparative continuous data with dichotomous data, we transformed logarithm odds ratios to effect size, assuming a normal underlying distribution [24] Quantitative synthesis was first conducted by comparing the d-dimer levels of CAP patients with various outcome. The between-study heterogeneity was evaluated by the $\chi2$-based Q statistics and $I^2$ test, and a significant heterogeneity was as $P<0.1$ [25] or $I^2>50\%$. When significant heterogeneity was observed, we would apply the random effects models for analysis. Otherwise, we would apply the fixed effects models. A sensitivity analysis was also conducted by sequential removal of each study. Here, we applied funnel plots as well as Egger's test [26] to assess publication bias. A two-sided P value of 0.05 was deemed as statistical significance.

The proportion patients with poor outcome correctly classified by the d-dimer levels as high risk was defined by dividing true-positive results by the sum of true-positive and false-negative results. The proportion of patients without poor outcome correctly classified by the d-dimer levels as low risk was defined by dividing true-negative results by the sum of true-negative and false-positive results. It was similar in concept to sensitivity and specificity although sensitivity and specificity are more appropriately reported at a particular time point in prognostic studies. Dose-response meta-analysis (DRMA) was conducted only for adjusted outcomes with more than 3 categories of exposures. When pooled effects had significant heterogeneity, and included more than 9 studies, subgroup analyses were carried out based on methodologies of d-dimer measurements, study design, location, sample size, risk of bias, number of categories and effect size type.

For network meta-analysis, we evaluated global inconsistency by fitting consistency and inconsistency model [27], and evaluated local inconsistency between direct and indirect estimates by using a node-splitting procedure [28]. In order to further quantify the d-dimer level of various outcome, we calculated the frequentist analogue of the surface under the cumulative ranking curve (SUCRA) for each outcome [29].

Data was analyzed using STATA Version 14.0 [30]. The network was evaluated using frequentist multivariate meta-analysis (commands *network meta* and *mvmeta*) in Stata 14.0. Besides, publication bias and sensitivity analysis were also conducted by STATA version 14.0.

**Table 1. Basic characteristics of included studies.**

| Author | Country | Year | Study design | primary outcome | Measured Assay | Sample size | Male (%) | Age | Comparison | Number of clinical centers |
|---|---|---|---|---|---|---|---|---|---|---|
| Agapakis [31] | Greece | 2010 | PR | Severity | Immunoturbidimetric assay | 108 | 61.1 | 65.11±8.34 | CAP VS healthy | Single |
| Arslan [32] | China | 2010 | RE | Severity | Latex immunoassay | 84 | 46.4 | 61.67 imm75 | CAP VS healthy | Single |
| Castro [45] | Spain | 2001 | PR | PE | ELISA | 101 | 46.5 | 46.23±12.19 | CAP VS PE | Single |
| Chalmers [33] | England | 2009 | PR | Mortality | Vitek ImmunoDiagnostic Assay System | 314 | 53.8 | N/A | Survivor VS No | Single |
| Chen [50] | China | 2020 | RE | Mortality | Not available | 179 | 66.5 | 65.0 (53.0–79.0) | Survivor VS non-survival | Single |
| Dai [51] | China | 2018 | RE | Mortality | Not available | 520 | 55.2 | N/A | Survivor VS non-survival | Single |
| Nastasijević [39] | Serbia | 2014 | RE | Severity and Mortality | Latex immunoassay | 129 | 59.7 | 64.8x immu | Severity, survival VS non-survival | Single |
| Duarte [56] | Portugal | 2015 | PR | Severity | Not available | 102 | 63.7 | 80.49v11.41 | No | Single |
| Mikaeilli [38] | Turkey | 2016 | PR | PE | Immunoturbidimetric method | 72 | 38.2 | 67.64±12.49 | PE VS CAP | Single |
| Ho [46] | Australia | 2013 | RE | PE | Not available | 472 | 48.4 | 61.15vaila6 | No | Single |
| Jin [35] | China | 2018 | PR | Severity | Immunoassay | 277 | 50.5 | 3.32 | CAP VS Control | Single |
| Kline [47] | USA | 2012 | PR | Severity | ELISA | 277 | 38.0 | 55.06 | PE VS Control | Multicenter - |
| Kobayashi [52] | Japan | 2016 | RE | Mortality | Not available | 3153 | 45.3 | 61.99vail28 | Survivors VS non-survival | Single |
|  |  |  |  |  |  |  |  |  |  | Single |
| Krykhtina [60] | Ukraine | 2019 | RE | CAP | Immunoturbidimetric method | 91 | 80 | 48.0 [33.0–61.0] | CAP VS Healthy | Single |
| Li [61] | China | 2017 | RE,CC | Severity | D-dimer assay kit | 302 | 57.3 | 8.10±1.80 | CAP VS Healthy | Single |
| Luo [48] | China | 2014 | PR | PE | ELISA | 57 | 50.9 | 61.36±10.70 | PE VS Non-PE | Single |
| Cerda-Mancillas [11] | Mexico | 2020 | PR | severity | fluorescence immunoassay | 52 | 71.6 ± 15 |  | Severity | Single |
|  |  |  |  |  |  |  |  |  | IMV VS Non-IMV |  |
|  |  |  |  |  |  |  |  |  | Vasopressor vs Non-vasopressor |  |
|  |  |  |  |  |  |  |  |  | Survivor VS non-survival |  |
| Marinkovic [37] | North Macedonia | 2016 | RE | Severity | Latex immunoassay | 192 | 58.9 | 53.97±17.71 | CAP VS PE | Single |
| Michelin [59] | Italy | 2008 | RE | Severity | D-dimer assay kit | 39 | - | 5.58 (2–174 months) | Severity of CAP | Single |
| Mikaeilli [38] | Iran | 2009 | RE | Mortality | ELISA | 60 | 35 | 47.12lityof | Survivor VS non-survival | Single |
| Milbrandt | USA | 2009 | PR | Severity and Mortality | Latex immunoassay | 939 | 51.4 | 69.20±15.80 | Survivor VS non-survival | Single |
| Ning Li [36] | China | 2018 | RE | Severity | immunoturbidimetric method | 96 | 46.9 | 17.81±3.48 | Severity | Single |
| Güneysel [34] | Greece | 2004 | PR | Severity | ELISA | 68 | 55.9 | 57.80±16.50 | healthy VS CAP VS severe CAP | Single |
| Paparoupa [49] | Germany | 2016 | RE | PE | D-Dimer Test Innovance from Siemens | 90 | 58 | 66.40±17.50 | CAP VS healthy | Single |
| Pereira [53] | Portugal | 2019 | PR | Mortality | Immunoturbidimetric assays | 107 | 65 | 62.00±15.70 | survival VS non-survival | Single |
| Pertseva [40] | Ukraine | 2019 | RE | Severity | ImmunoturbIdimetric assays | 73 |  | 54.0 [37.0–63.0] | Severity | Single |

*(Continued)*

**Table 1.** (Continued)

| Author | Country | Year | Study design | primary outcome | Measured Assay | Sample size | Male (%) | Age | Comparison | Number of clinical centers |
|---|---|---|---|---|---|---|---|---|---|---|
| Querol-Ribelles [41] | Spain | 2004 | PR | Severity and Mortality | Automated latex assay | 302 | 74.8 | 73.00 | different clinical outcome | Single |
| Salluh [54] | Brazil | 2011 | PR | Mortality | Coagulation A Not available lyzer | 90 | 44.4 | 73.5(57.7–83) | suvivors VS non-survival, complications VS non-complications | Single |
| Shilon [42] | Israel | 2003 | PR | Severity | Miniquant D-dimer assay | 68 | 40 | 67.00±20.80 | CAP VS healthy | Single |
| Snijders [43] | Netherlands | 2012 | PR | Severity | ELISA | 147 | 53.7 | 63.1Aityn | CAP VS healthy | Single |
| Yende [55] | USA | 2011 | PR | Mortality | Latex immunoassay | 893 | 51.2 | 68.7 (15, 73) | Survivors VS non-survival | Multicenter |
| Zhang [58] | China | 2015 | RE | PE | Immunoturbidimetry | 139 | 47.48 | 70.73 | PE VS Non-PE | Single |

Data are presented as n (%), mean±standard deviation or median (interquartile range)

PE = pulmonary embolism. ELISA = enzyme-linked immunosorbent assay, PR = prospective cohort study, RE = retrospective cohort study, CC = case-control study,

IMV = invasive mechanical ventilation

## Result

Nine-hundred and twenty-five articles were retrieved from databases, of which 32 studies with a total of 9,593 patients were eventually included (Fig 1). No additional relevant articles were identified in the bibliographies of the original articles. The characteristics of the included studies are listed in Table 1.

## Characteristics of included studies

D-dimer levels were reported in 14 studies [11, 31–43] associated with severity, 6 [44–49] with PE, 12 [11, 38, 39, 41–43, 45, 50–54] with mortality and 3 [38, 39, 41] with invasive mechanical ventilator. All included studies were observational. Two [47, 55] of them were multicenter and eleven [31, 33, 34, 37, 39, 42, 43, 45, 48, 54, 55] were prospective observational studies. Five studies [32, 33, 38, 39, 41] reported d-dimer levels of different severity by class I to class V so that we combined means and standard deviation (SD) into two groups (severe CAP VS Non-severe CAP) by StatsToDo. Three studies reported dichotomous outcomes according to different categories of d-dimer without adjusted effect size and confidence interval (or standard error) compared to reference category. Therefore, dose-response meta-analysis is unavailable.

## D-dimer measurement

Recommended thresholds of different d-dimer assays varied widely across studies. Cut-off values were determined as 500 or 1000 ng/mL in 7 studies [33, 41, 48, 53, 56–58]. ELISA method was used in 7 studies [34, 35, 38, 43, 45, 48, 59]; immunoturbidimetric methods were used in 7 studies [31, 36, 37, 40, 44, 53, 60] and quantitative latex assay were launched in six studies [32, 37, 39, 41, 55, 57]. The rest of included studies only reported D-dimer kit, automatic analytical instrument, but not revealed their assay methods.

## Assessment of methodological quality

QUIPS tool had been used to assess the quality of included studies (Table 2). Among all the 32 studies, 20 were high risk of bias [11, 31, 32, 34, 35, 37–42, 44, 49, 54–56, 58–61], 2 studies

**Table 2. Quality assessment of individual studies using the QUIPS instrument.**

| Study | Study Participation | Study Attrition | Prognostic Factor Measurement | Outcome Measurement | Study Confounding | Statistical Analysis and Reporting | Overall Assessment |
|---|---|---|---|---|---|---|---|
| Agapakis 2010 | L | L | M | L | H | L | H |
| Arslan 2010 | H | H | L | L | M | L | H |
| Castro 2001 | L | M | L | M | L | L | M |
| Chalmers 2009 | L | L | L | L | M | L | M |
| Dai 2018 | L | L | L | M | L | L | M |
| Nastasijević 2014 | H | M | L | L | H | L | H |
| Mikaeilli 2016 | H | M | L | L | H | L | H |
| Ho 2013 | L | L | M | L | L | L | M |
| Jin 2018 | L | M | M | L | H | L | H |
| Kline 2012 | L | L | L | L | M | L | M |
| Kobayashi 2016 | L | L | L | M | M | L | M |
| Krykhtina 2019 | H | L | L | L | H | L | H |
| Li 2017 | H | L | L | L | H | L | H |
| Cerda-Mancillas 2020 | H | L | L | L | M | H | H |
| Marinkovic 2016 | H | H | L | L | H | M | H |
| Michelin 2008 | H | L | L | L | M | L | H |
| Mikaeilli 2009 | M | H | L | L | H | M | H |
| Milbrandt 2009 | L | L | L | L | M | L | M |
| Oziem 2004 | H | H | L | M | H | L | H |
| PaparoMpa 2016 | H | H | L | L | M | L | H |
| Pertseva 2019 | H | L | L | L | H | L | H |
| Pereira 2019 | L | L | L | L | L | L | L |
| Querol-Ribelles 2004 | L | L | L | L | H | L | H |
| Salluh 2011 | H | L | L | L | H | L | H |
| Shilon 2003 | H | L | L | L | H | L | H |
| Snijders 2012 | L | L | L | L | L | L | L |
| Yende 2011 | L | L | L | L | H | L | H |
| Zhang 2016 | H | L | L | L | H | L | H |

Low: Low risk of bias; Moderate: Moderate risk of bias; High: High risk of bias. U: Unclear.

were evaluated as low risk of bias [43, 53] and the remaining 10 studies had medium risk of bias. The number of individual domains rated as high risk ranged from 0 to 3.

## The association between D-dimer level and clinical outcomes

**Severity.** We excluded one study [59] for quantitative synthesis because it simply defined severe CAP as pneumonia with pleural effusion. Twelve studies [11, 31, 32, 34, 35, 37–43] including 1,394 participants reported the plasma d-dimer level of both severe CAP and non-severe CAP patients. Pooled effect size (ES) suggested d-dimer level was significantly higher in severe CAP patients than non-severe CAP patients with great heterogeneity (SMD = 1.21 95% CI 0.87–1.56, $I^2$ = 86.8% p = 0.000) (Fig 2) and pooled result with adjusted OR [36, 43] (OR = 1.07 95%CI 1.01–1.13, $I^2$ = 60.2% p = 0.113) was consistent with continuous data (S1 Fig). By subgroup analysis, conversion of median and quartiles, combination of mean and SD,

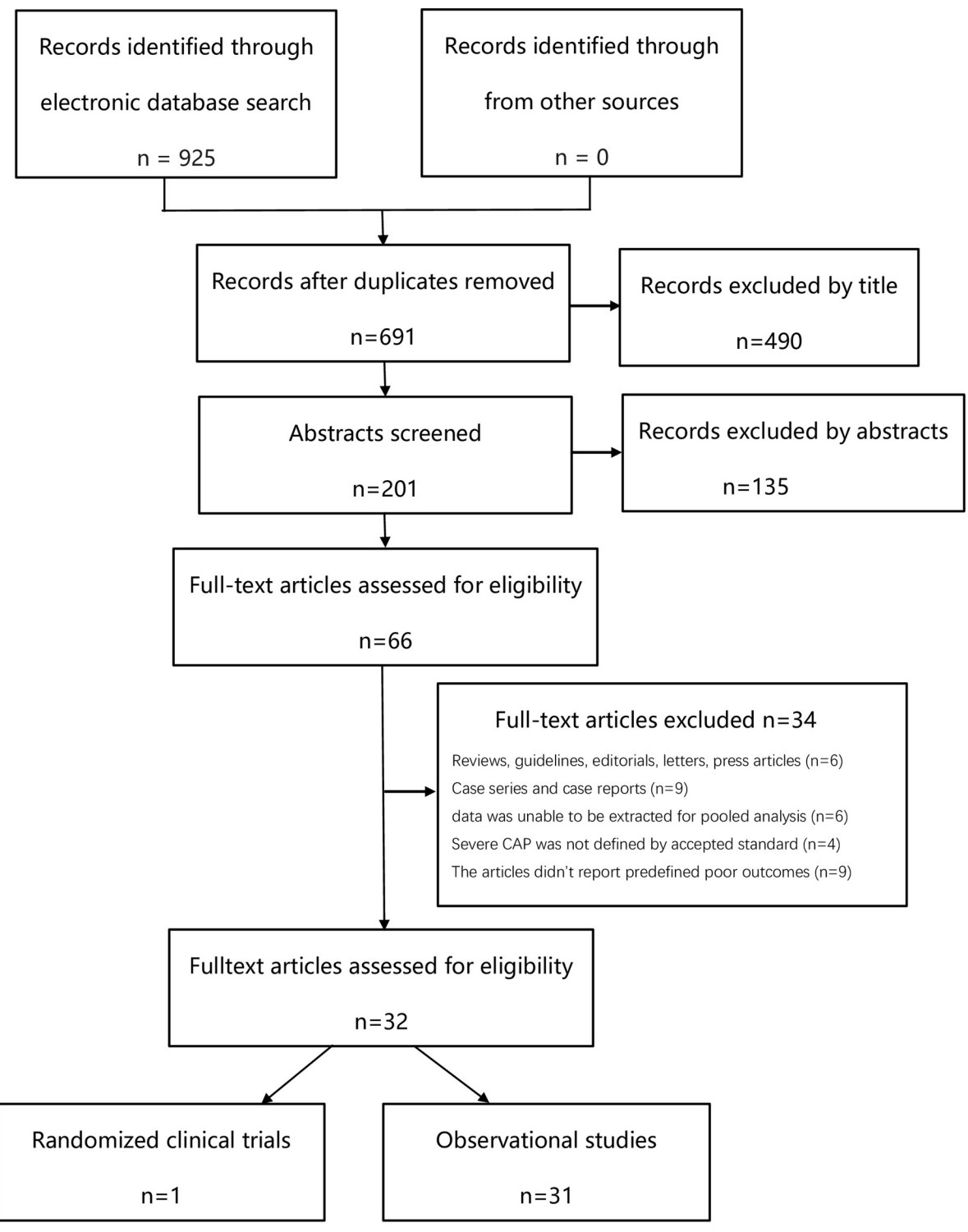

Summary of the literature search and inclusion process

**Fig 1. Preferred Reporting Items for Systematic Reviews and Meta-Analyses (PRISMA) flow diagram for study identification and selection.**

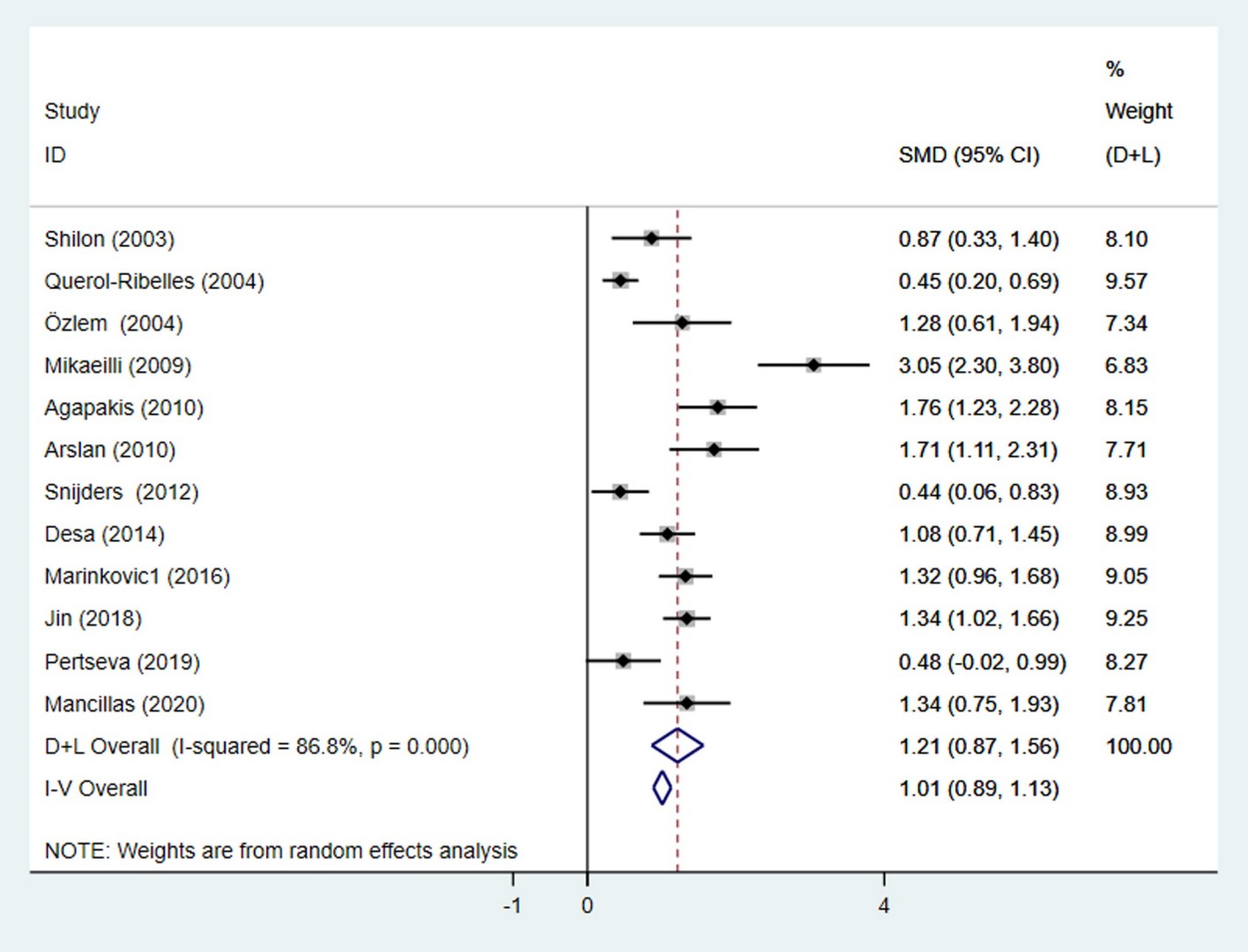

**Fig 2. Forest plot of D-dimer in severe CAP versus non-severe CAP patients.** The size of the square is proportional to study-specific statistical weights, horizontal lines represent 95% confidence interval (CI) and diamonds represent summary measures of association. SMD, standardized mean difference; ES, effect size. CAP, community-acquired pneumonia.

methodologies of d-dimer measurements, high risk of bias and tools of d-dimer measurement did not contribute to heterogeneity significantly.

Sensitivity analysis of pooled ES of continuous data suggested that no study contribute much to the pooled estimate as our findings remained consistent (S2 Fig). Funnel plots were used to assess publication bias. We detected asymmetry in the funnel plot which was further ascertained by Egger's test (P = 0.031; Fig 3), suggesting the presence of publication bias.

**Mortality.** Mortality associated with CAP was an important clinical outcome. 12 articles had been included to make quantitative analysis. Among them, 10 articles [11, 38, 39, 41–43, 50, 52–54] including 4,117 participants reported the plasma D-dimer level of survivors and non-survivors with CAP, which confirmed significant elevation of D-dimer level in non-

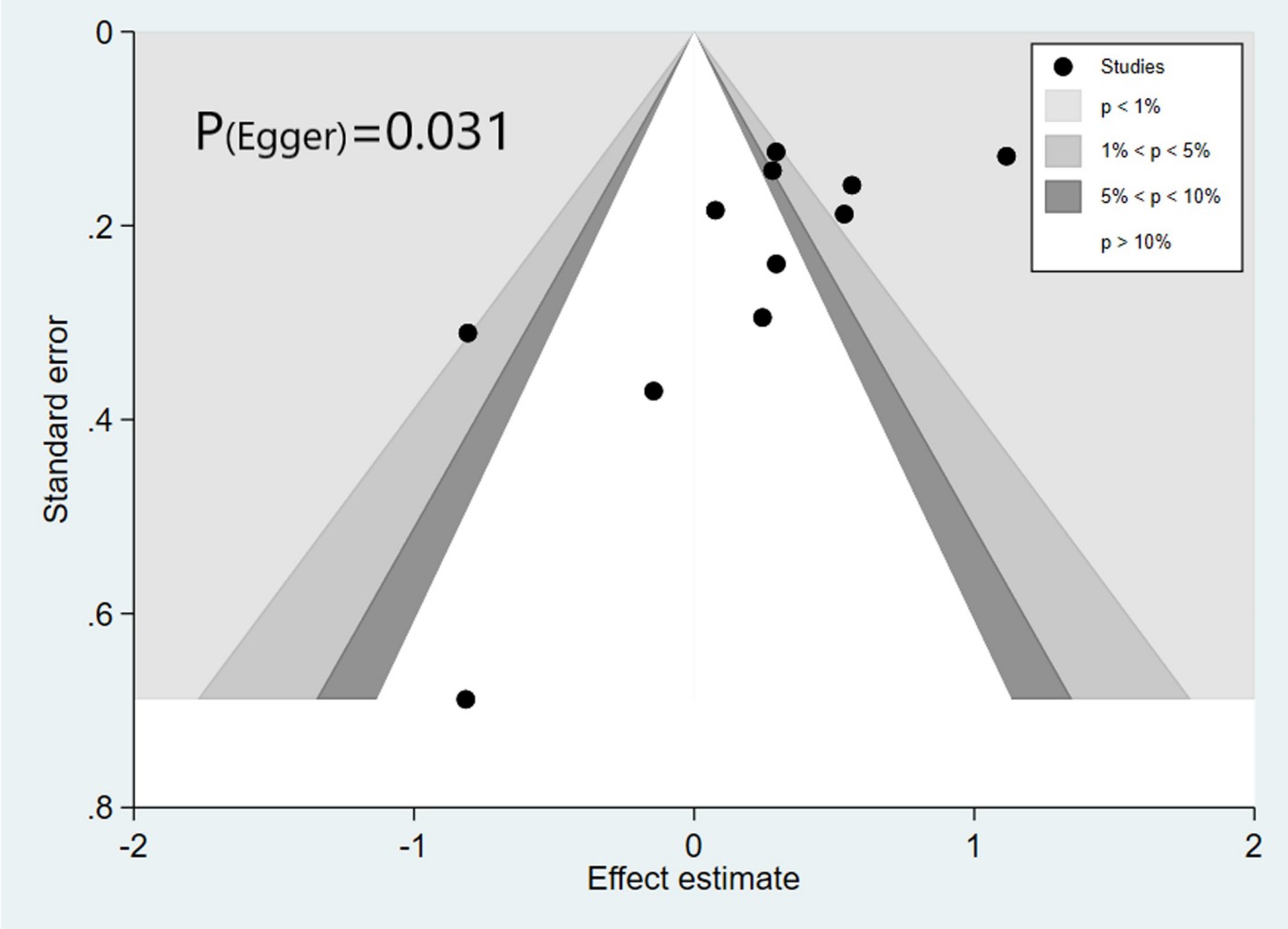

**Fig 3. Funnel plot with Egger's test for d-dimer levels and severity.** Unadjusted effect estimates from individual studies were plotted against their standard error. Solid and dashed lines represent the summary effect estimate and its 95% confidence intervals for different values of the standard error, respectively. Egger's test estimated bias: p = 0.031. (A) Funnel plot assessing publication bias in RCTs investigating the effectiveness of different types of respiratory PPE against clinical (influenza-like illness and clinical respiratory illness) or laboratory-confirmed outcomes (influenza or other viral or bacterial respiratory infections); Harbord's estimated bias coefficient: -0.59; p = 0.592.

survivors (SMD = 0.90 95%CI 0.62–1.17, $I^2$ = 59.4% p = 0.008) (Fig 4). In spite of great heterogeneity, the pooled ES was robust by sensitivity analysis (S3 Fig). Pooled results of adjusted ORs from 4 studies [51–53, 55] have confirmed the prognostic value of increased D-dimer (OR = 2.23 95%CI 1.15–3.31, $I^2$ = 0.0% p = 0.791) (S4 Fig).

The presence of asymmetric distribution of funnel plots suggested that there might be publication bias in pooled result of continuous data (S5 Fig). Paradoxically, qualitative analysis by Egger's test did not indicate publication bias (p = 0.168).

**Pulmonary embolism.**   Pooled effect based on continuous data from three studies suggest a significant elevation in CAP patients with PE than those without PE (SMD = 0.75 95%CI 0.11–1.38, $I^2$ = 84.5% p = 0.002) (S6 Fig). As there were only three enrolled studies, the publication bias and subgroup analysis were unavailable. However, data from a series of studies could be converted to the form of fourfold table of diagnostic test to demonstrate the prognostic value

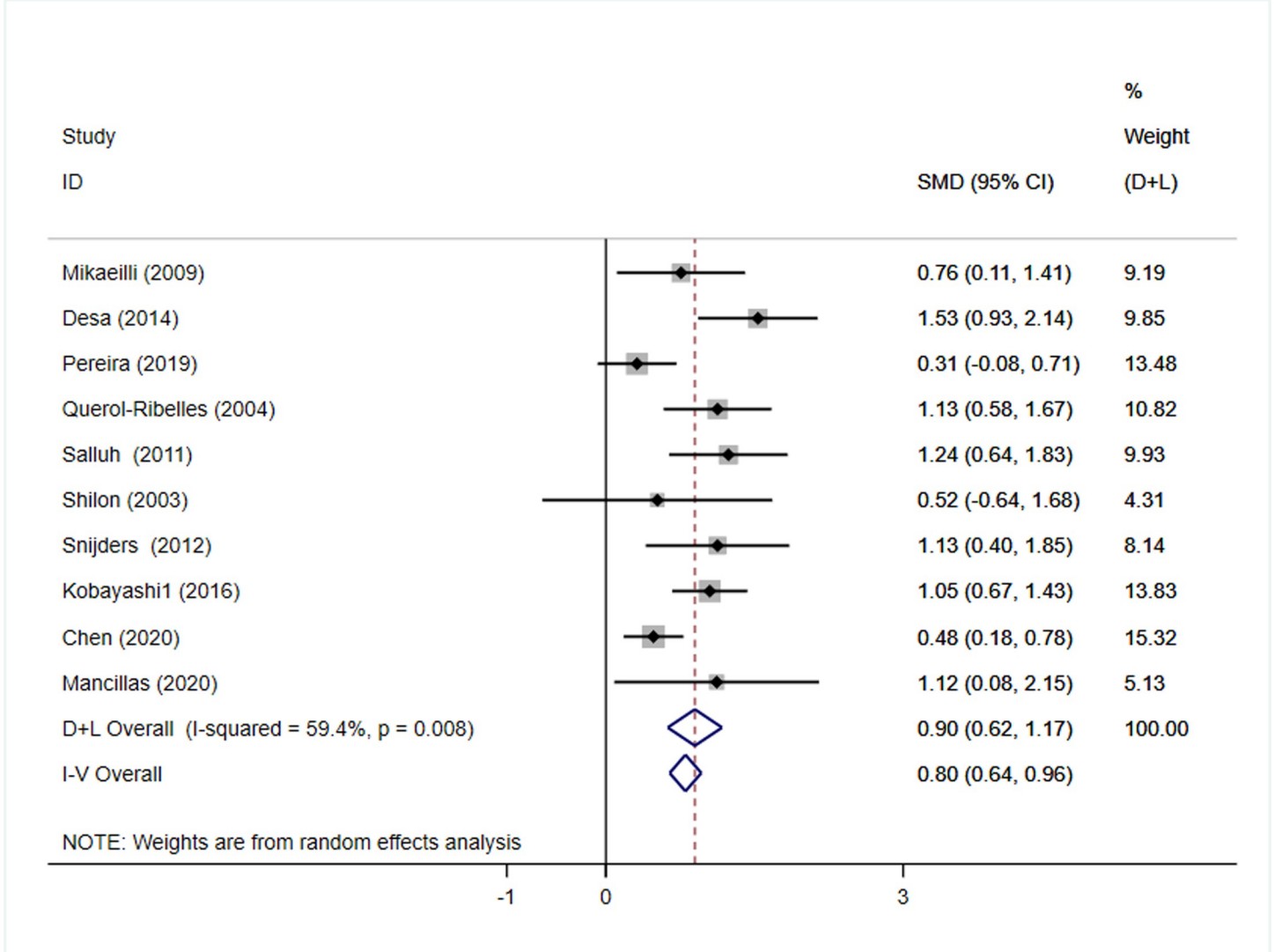

**Fig 4. Forest plot of D-dimer in non-survivors versus survivors with CAP.** There was a significant elevation of D-dimer level in non-survivors with great of heterogeneity between trials. CAP, community-acquired pneumonia.

of D-dimer in high-risk population of PE. A diagnostic meta-analysis assessment was made to further investigate the prognostic role of D-dimer. The summary sensitivity was 0.74 (95%CI, 0.50–0.89), with significant heterogeneity (P = 0.0001, $x^2$ = 26.86, $I^2$ = 81.4%) (S7 Fig). The summary specificity was 0.82 (95%CI, 0.41–0.97), and the pooled estimation showed significant heterogeneity (P = 0.0000, $x^2$ = 498.29, $I^2$ = 99.0%) (S7 Fig). HSROC curve showed potential prognostic value of d-dimer levels for patients at high risk of mortality with CAP (Fig 5).

**Others.** Pooled effects suggested patients with CAP had a higher level of d-dimer compared to healthy participants (SMD = 0.88, 95%CI 0.54–1.22, $I^2$ = 57.9%, p = 0.037) (S8 Fig) and D-dimer was significantly further elevated in patients requiring invasive mechanical ventilator (SMD = 1.01, 95%CI 0.69–1.33, $I^2$ = 0.0%, p = 0.815) (S9 Fig).

**Network analysis.** We pooled effects of different outcomes by network meta-analysis of frequentist statistics (S10 Fig) and loop inconsistency test suggested significant heterogeneity between direct and indirect comparisons (S11 Fig). The results provide evidence that there

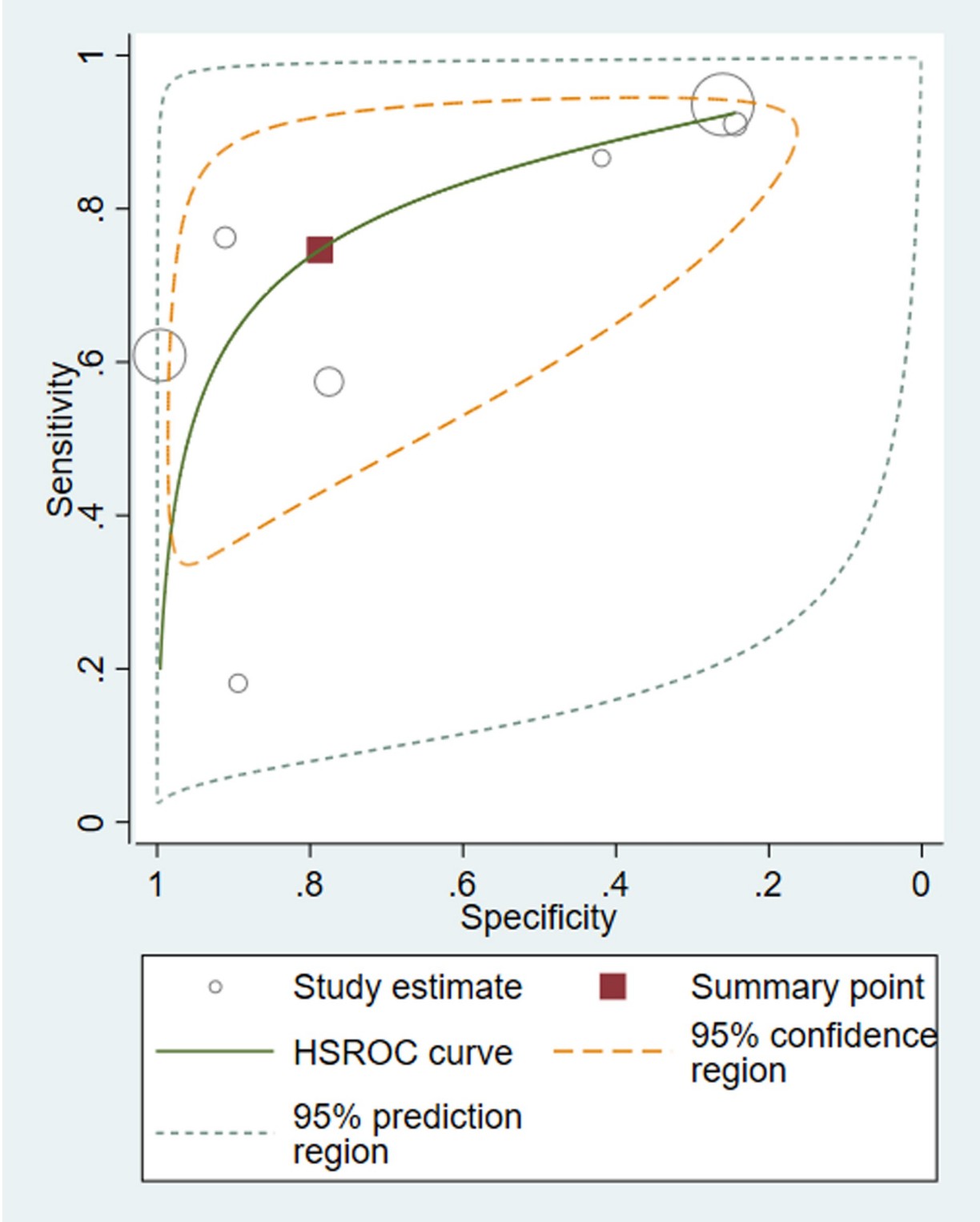

**Fig 5. HSROC for d-dimer levels and occurrence of pulmonary embolism in patients with CAP.** HSROC, hierarchical summary receiver operating characteristic curve.

was significant elevation of d-dimer levels in CAP patients with poor outcomes than general CAP patients. But wide 95% prediction interval cross null value (0) reminded us potential heterogeneity of included studies (Fig 6). D-dimer levels weren't significantly different among poor outcomes although the SUCRA statistic showed that d-dimer level in CAP patients requiring mechanical ventilators ranked first, followed by non-survivors, severe patients and patients with PE. (S12 Fig). Funnel plots suggested potential publication bias based on its slight asymmetry (S13 Fig).

**Quality of evidence.**   Most of included studies are retrospective observational studies and had high or moderate risk of bias. Great heterogeneity existed in pooled effects and contributed to inconsistency of evidence. Based on GRADE system and the above considerations, the quality of evidence of our study should be low or very low.

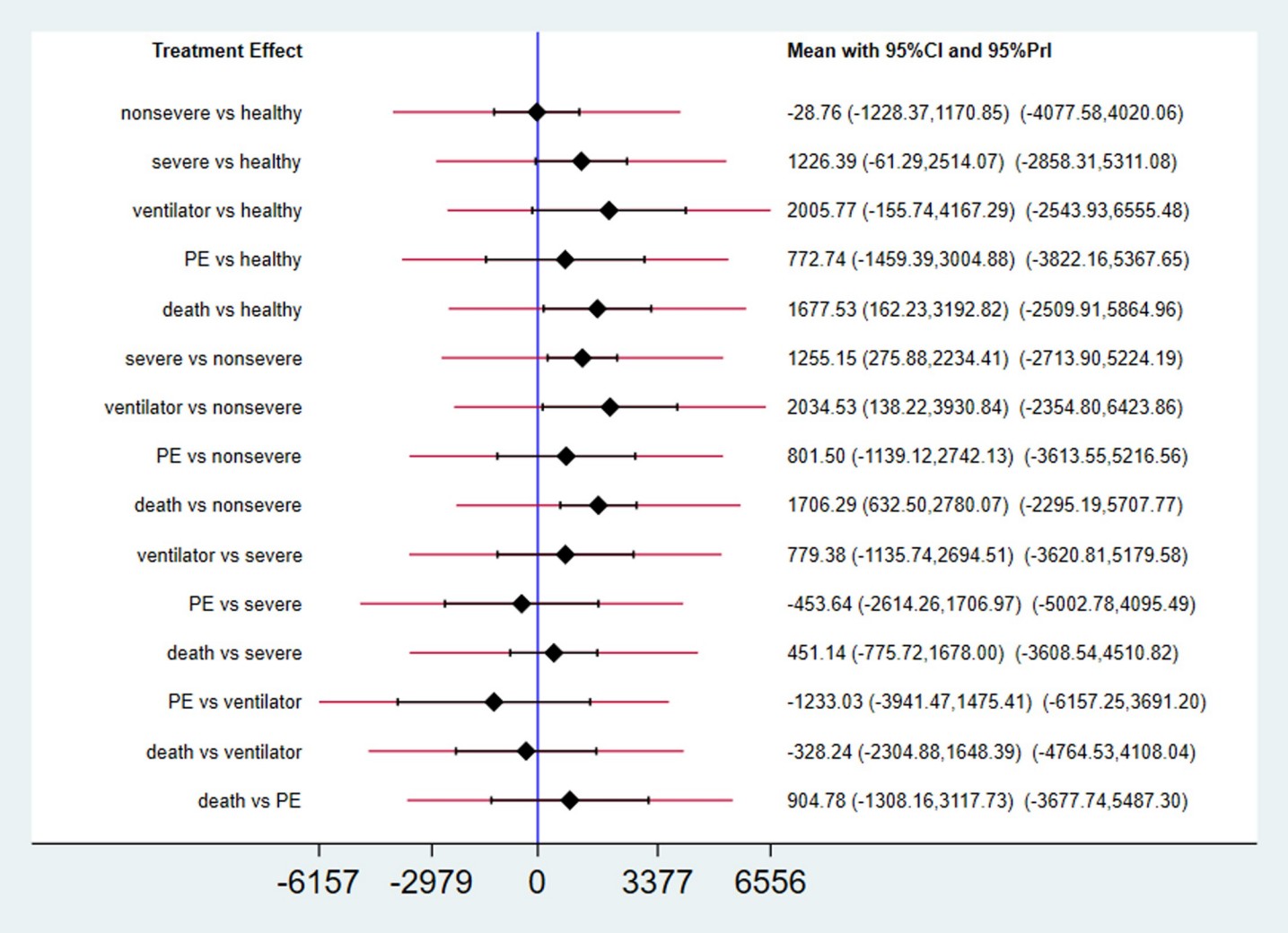

**Fig 6. Forest plot of network meta-analysis of SMD of d-dimer levels of patients with different outcomes.** The red lines show prediction interval of future research. PrI = prediction interval. SMD = standardized mean difference.

## Discussion

Our study is the first meta-analysis ever published about prognostic value of D-dimer in patients with CAP. It shows that elevated d-dimer level is significantly associated with CAP severity, mortality and PE occurrence in general analysis regardless their age, gender, race and region and the type of assay of d-dimer measurement. With the aid of method of diagnostic meta-analysis, we confirmed prognostic value of d-dimer for high-risk population of PE in patients with CAP. Although we were unable to establish a firm evidence on the independent prognostic value of d-dimer levels, our results on d-dimer levels were consistent and robust. Network meta-analysis further confirmed the evidence of conventional meta-analysis and suggested d-dimer levels were not significantly different in various poor outcomes. This biomarker may be helpful in the early identification of patients with high risk of poor outcomes to make special therapeutic strategy as soon as possible.

Previous studies reported that patients with elevated D-dimer levels were more likely to suffer from thromboembolism [62], digestive cancer [63], traumatic brain injury [64], and aortic dissection [65], which are associated with coagulation disorders. However, the pathophysiology of D-dimer elevation in CAP is only partially understood. Indeed, D-dimer elevation has also been observed in children and adults without any symptoms of pneumonia. Elevation of D-dimer had been reported to be correlated with several inflammatory and coagulation factors, including C-reaction protein, procalcitonin, IL-6, prothrombin time (PT), activated partial thromboplastin time (APTT) and thrombin time (TT) [7, 9, 66, 67]. Besides, several studies showed some patients with high d-dimer levels who died from severe CAP did not present obvious disorders of coagulation function [31].

As an unconventional detection index, the value of d-dimer varied widely across different studies, resulting in great unmanageable heterogeneity in pooled effects. This is partly because of diverse measuring apparatus and assays. The primary outcome is problematic for time-to-event analyses, particularly if studies have short follow-up and significant censoring. On the other hand, it is difficult to make subgroup analysis because of limited number of included studies and insufficiently reported data to make stratification. D-dimer is only reported as a confounding factor in most of included studies so the data from them is unable to support to make dose-response meta-analysis. Moreover, given the heterogeneity in study designs and data reporting, as well as the lack of availability of individual patient data, meta-analysis of hazard ratios was not feasible. There is uncertain statistical bias in combine means and SDs into one group and estimating the sample mean and SD from the sample size, median, range and/or interquartile range.

## Conclusion

This study found that the prognostic ability of d-dimer to predict multiple poor outcomes among patients with CAP. But it is difficult to distinguish high-risk populations of different poor outcomes according to d-dimer levels. Additional, more rigorously structured research appears to be needed to better quantify the association of d-dimer levels with poor outcomes in patients with CAP and to demonstrate clinical utility.

## Supporting information

**S1 Fig. Forest plot of pooled ORs of D-dimer in severe CAP versus non-severe CAP patients.** OR, odds ratio. CAP, community-acquired pneumonia.
(TIF)

**S2 Fig. Sensitivity analysis of the individual trials on the results for plasma D-dimer level associated with severity.**
(TIF)

**S3 Fig. Sensitivity analysis of the individual trials on the results for plasma D-dimer level associated with mortality.**
(TIF)

**S4 Fig. Forest plot of pooled ORs of D-dimer in survivors versus non-survivors with CAP.** CAP, community-acquired pneumonia.
(TIF)

**S5 Fig. Funnel plot with Egger's test for association between d-dimer levels and mortality.**
(TIF)

**S6 Fig. Forest plot of D-dimer in CAP patients with or without PE.** CAP, community-acquired pneumonia. PE, pulmonary embolism.
(TIF)

**S7 Fig. Forest plot of pooled sensitivity and pooled specificity for d-dimer levels and occurrence of pulmonary embolism.**
(TIF)

**S8 Fig. Forest plot of D-dimer in CAP patients versus healthy participants.** SMD, standardized mean difference; CAP, community-acquired pneumonia.
(TIF)

**S9 Fig. Forest plot of D-dimer in common CAP patients versus CAP patients requiring invasive mechanical ventilators.** SMD, standardized mean difference; CAP, community-acquired pneumonia.
(TIF)

**S10 Fig. The network meta-analysis of available comparisons of d-dimer levels of patients with various outcomes.** The line width is proportional to the number of trials performed between two outcomes. Circle size is proportional to the total number of patients for each clinical outcome in the network.
(TIF)

**S11 Fig. The examination of loop inconsistency.** RoR: The rate ratio of logarithms of two ORs of direct and indirect comparisons.
(TIF)

**S12 Fig. Results of network rank test and the surface under the cumulative ranking curve (SUCRA).**
(TIF)

**S13 Fig. Comparison-adjusted funnel plot for d-dimer levels of patients with various clinical outcomes.**
(TIF)

**S1 Table. Extracted data for meta-analyses.**
(XLSX)

**S1 File.**
(DOC)

## Author Contributions

**Conceptualization:** Kaiyu Zhou, Hongyu Duan, Yimin Hua, Yifei Li.

**Data curation:** Jiawen Li, Kaiyu Zhou, Hongyu Duan, Yimin Hua, Yifei Li.

**Formal analysis:** Kaiyu Zhou, Hongyu Duan, Yifei Li.

**Funding acquisition:** Hongyu Duan, Yifei Li.

**Investigation:** Jiawen Li, Hongyu Duan, Peng Yue, Yimin Hua, Yifei Li.

**Methodology:** Jiawen Li, Kaiyu Zhou, Hongyu Duan, Peng Yue, Xiaolan Zheng, Yimin Hua, Yifei Li.

**Project administration:** Yimin Hua, Yifei Li.

**Resources:** Jiawen Li, Xiaolan Zheng.

**Software:** Jiawen Li, Hongyu Duan, Xiaolan Zheng, Lei Liu, Hongyu Liao, Yifei Li.

**Supervision:** Jinlin Wu, Jinhui Li, Yimin Hua.

**Validation:** Jiawen Li, Lei Liu, Hongyu Liao, Yifei Li.

**Visualization:** Jiawen Li, Lei Liu, Hongyu Liao, Yifei Li.

**Writing – original draft:** Jiawen Li, Yifei Li.

**Writing – review & editing:** Yimin Hua, Yifei Li.

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
