## [Decision Letter · Decision Letter 0]

16 Feb 2021

PONE-D-20-17727

Assessment the value of D-dimer in predicting various clinical outcomes following CAP: A network meta-analysis

PLOS ONE

Dear Dr. Li,

Thank you for submitting your manuscript to PLOS ONE. After careful consideration, we feel that it has merit but does not fully meet PLOS ONE’s publication criteria as it currently stands. Therefore, we invite you to submit a revised version of the manuscript that addresses the points raised during the review process.

We look forward to receiving your revised manuscript.

Kind regards,

Cho Naing, MBBS, PhD, FRCP

Academic Editor

PLOS ONE

Journal Requirements:

Additional Editor Comments:

The methodology is insufficient.

Moreover, the selection criteria is not clear. It will be more meaningful and well focused to include randomized controlled trials, rather than non-randomized studies

The assumptions required for a network- meta analysis are not addressed in the methods, and a lack of reporting them in the results.

Thank you

Reviewers' comments:

Reviewer's Responses to Questions

**Comments to the Author**

1. Is the manuscript technically sound, and do the data support the conclusions?

Reviewer #1: Partly

Reviewer #2: Yes

2. Has the statistical analysis been performed appropriately and rigorously? 

Reviewer #1: I Don't Know

Reviewer #2: Yes

3. Have the authors made all data underlying the findings in their manuscript fully available?

Reviewer #1: No

Reviewer #2: Yes

4. Is the manuscript presented in an intelligible fashion and written in standard English?

Reviewer #1: No

Reviewer #2: Yes

5. Review Comments to the Author

Reviewer #1: The authors Li et al presented their network meta-analysis on the utility of D-dimer as biomarkers of community-acquired pneumonia. While the paper has merit, this is hardly suitable for publication at this stage.

1. English used are mostly non-standard, contain a lot of errors and may create confusion. This alone warrants a major revision. I strongly urge a scientific English editing before going to the next review round. I would not go for the next round of review unless the authors provided evidence of scientific english editing. I have limited myself to review up to methodology section and a few tables only and hold it on until sufficient english editing is done for reviewing further.

2. The authors claimed of using Cochrane Handbook for their analysis. Which Cochrane Handbook was used? I understand there is a diagnostic test accuracy handbook currently in process of development. I dont think the intervention handbook is appropriate for this study.

3. Search strategy. Search is quite outdated (28 Oct 2019) which practically excluded an important aetiology, SARS-CoV-2, unless the study purposely wanted to exclude that.

4. Inclusion criteria. Dichotomy between case-control and cohort studies could be confusing. I think, in any case, the study should be comparing between different groups of diseases (CAP, healthy, PE, etc) using the particular marker of interest (D-dimer). Studies could be either cross-sectional or cohort (observation over particular timeframe).

5. Inclusion criteria 7 and 8 are confusing when they dichotomize case-control and cohort. Criterias in 7 should be applicable into 8 and vice versa. Inclusion criteria 10 is unclear.

6. Does "repeat publication" means "duplication"? If so, this is normally understood as part of the screening process.

7. What do the authors mean by "This metaanalysis would prefer to include the studies deemed moderate to high methodological quality which should be scored at least 5 stars."? Does this mean studies were excluded if they dont qualify for this?

8. Table 1. All acronyms: PR, CR, PE, etc should be explained at the bottom of the table.

9. Table 1 in column whether comorbidities were excluded. Some studies were "No". Since certain comorbidities are to be excluded. This warrants further detailing of what comorbidities were included.

10. Table 1. What does area refer to? Earlier, 4 studies were reported as multi-center, but none in "area" column indicated that.

Reviewer #2: The manuscript is well prepared and data analysis was done thoroughly.

The following are a few minor suggestions for improvement.

Title

“Assessing the value” or “assessment of the value of” or “Value of D-dimer in predicting various clinical outcomes following CAP: A network meta-analysis

Abstract

Page 9: line 3: The role of D-dimer in predicting clinical outcomes is debating in many areas including the severity of community-acquired pneumonia (CAP), in hospital mortality and the risk of pulmonary embolism (PE). So that We aimed to carry out a meta-analysis to identify the role of d-dimer in predicting clinical outcomes which are associated CAP.

Page 9; line 11: Data “was” analyzed

Page 9: line 15: more suitable replacement for “hospital dead cases” and “PE attack patients”

Page 9: line 20: to explore the distinguished role

Page 9: line 26: PE high risk population � high risk population of PE

Background

Page 11

Line 1: one of the most common respiratory diseases

Line 4: CAP patients would progress into severe CAP

Line 5: However, for a large proportion of CAP cases who visit emergency department could receive medical treatment without hospitalization.

Line 7: critical for accurate treatment and to set setting up an appropriate care strategy

To find reference 6

Line 12: procalcitonin [9, 10] had been reported with a great sensitivity for predicting the prognosis of CAP

Line 14: Nevertheless, their specificities are not good enough to distinguish the various clinical

outcomes of CAP. (reference?)

28th Jan 2021Nevertheless, their specificities are not good enough to distinguish the various clinical outcomes of CAP.

Page 12

Line 21: The inclusion criteria were as followings: should be “as follows”.

Page 13

Line 21: Besides, “The quality” of predictive and diagnostic study’s. “the quality”

Page 15

Line 12: The network will be evaluated using frequentist multivariate meta-analysis. Should be “the network was evaluated”

Page 16

Line 4: after carefully evaluated or after careful evaluation.

Line 10: In addition, patients with diarrhea and congenital heart diseases were excluded in 3 studies of children. (not clear)

Page 17

Line 25: ORs had been pooled to identify.

Data analysis and interpretation was carried out properly.

Page 21 line 17

elevated D-dimer is an available supplement to PSI and CURB-65, which are the most widely used points-scoring system to assess severity of pneumonia. [50] --

what is noted from reference 50 is "The coagulation system is often activated in CAP, and development of thrombocytopenia (platelet count, <100,000 cells/mm3) is also associated with a worse prognosis [86, 90–92]."-

Is it quite certain with this reference only to use "is an available supplement to PSI and CURB-65 with this reference only "? thank you.

6. PLOS authors have the option to publish the peer review history of their article (what does this mean?). If published, this will include your full peer review and any attached files.

Reviewer #1: No

Reviewer #2: No

---

## [Author Response · Author response to Decision Letter 0]

7 May 2021

Dear editor,

We could not find any places in the system to update the finical statement, so that we typed our newest finical statement here. We confirmed the financial statement as the followings: “This work was supported by grants from the National Natural Science Foundation This work was supported by grants from the National Natural Science Foundation of China awarded to YL (81700360) and HD (81971457, 81602817). The funding source had no role in the design and conduct of the study; collection, management, analysis, and interpretation of the data; preparation, review, or approval of the manuscript; and decision to submit the manuscript for publication.”

Reviewer #1:

1. English used are mostly non-standard, contain a lot of errors and may create confusion. This alone warrants a major revision. I strongly urge a scientific English editing before going to the next review round. I would not go for the next round of review unless the authors provided evidence of scientific english editing. I have limited myself to review up to methodology section and a few tables only and hold it on until sufficient english editing is done for reviewing further.

Response: We are sorry for non-standard English. We have make scientific English editing in revised version. Thanks a lot.

2. The authors claimed of using Cochrane Handbook for their analysis. Which Cochrane Handbook was used? I understand there is a diagnostic test accuracy handbook currently in process of development. I dont think the intervention handbook is appropriate for this study.

Response: In 2019, the Cochrane Prognosis Methods Group has funded to make a guide to systematic review and meta-analysis of prognostic factor studies, which was published in BMJ. Therefore, we revised our articles according to this guide instead of Cochrane Intervention Handbook. Thanks for your advice.

3. Search strategy. Search is quite outdated (28 Oct 2019) which practically excluded an important aetiology, SARS-CoV-2, unless the study purposely wanted to exclude that.

Response: We updated our study to end of March 2021. In our opinion, COVID-19 is not a standard community-acquired pneumonia (CAP) with absolute different disease process and therapeutical strategies. SARS-Cov-2 is not one of pathogens of CAP based on current consensus, although it may be CAP pathogen in future. We have stated in our article why we didn’t include studies associated with COVID-19.

4. Inclusion criteria. Dichotomy between case-control and cohort studies could be confusing. I think, in any case, the study should be comparing between different groups of diseases (CAP, healthy, PE, etc) using the particular marker of interest (D-dimer). Studies could be either cross-sectional or cohort (observation over particular timeframe).

Response: We are sorry to make inclusion criteria confusing. We have remade our inclusion criteria according to modification of CHARMS (checklist for critical appraisal and data extraction for systematic reviews of prediction modelling studies) for prognostic factors (CHARMS-PF). Cross-sectional or and cohort studies were included according to “a guide to systematic review and meta-analysis of prognostic factor studies”. 

5. Inclusion criteria 7 and 8 are confusing when they dichotomize case-control and cohort. Criterias in 7 should be applicable into 8 and vice versa. Inclusion criteria 10 is unclear.

Response: We have advised as stated. Thanks a lot.

6. Does "repeat publication" means "duplication"? If so, this is normally understood as part of the screening process

Response: We have replaced "repeat publication" with “duplication” and made it as part of the screening process. Thanks for your advice.

7. What do the authors mean by "This metaanalysis would prefer to include the studies deemed moderate to high methodological quality which should be scored at least 5 stars."? Does this mean study were excluded if they dont qualify for this?

Response: According the guide, we didn’t exclude any study due to high risk of bias. We regarded methodological quality as one factor to make subgroup analysis.

8. Table 1. All acronyms: PR, CR, PE, etc should be explained at the bottom of the table.

Response: We added explanation of all acronyms at the bottom of the table. Thanks.

9. Table 1 in column whether comorbidities were excluded. Some studies were "No". Since certain comorbidities are to be excluded. This warrants further detailing of what comorbidities were included.

Response: The data associated with comorbidity is insufficient to make further analyses as d-dimer is not a main research topic in most included studies. Therefore, we didn’t include comorbidities in Table 1 in revised version.

10. Table 1. What does area refer to? Earlier, 4 studies were reported as multi-center, but none in "area" column indicated that. 

Response: We have replaced area with country and the multi-center studies were made in several hospital of the same country. Through careful screening again, only 2 studies meet the criteria of multicenter studies in our revised version.

Reviewer #2

Title

“Assessing the value” or “assessment of the value of” or “Value of D-dimer in predicting various clinical outcomes following CAP: A network meta-analysis. 

Response: We have revised our title as you suggest. Thanks for your advice.

Abstract

(1) Page 9: line 3: The role of D-dimer in predicting clinical outcomes is debating in many areas including the severity of community-acquired pneumonia (CAP), in hospital mortality and the risk of pulmonary embolism (PE). So that We aimed to carry out a meta-analysis to identify the role of d-dimer in predicting clinical outcomes which are associated CAP.

(2) Page 9; line 11: Data “was” analyzed

(3) Page 9: line 15: more suitable replacement for “hospital dead cases” and “PE attack patients”

(4) Page 9: line 20: to explore the distinguished role

(5) Page 9: line 26: PE high risk population � high risk population of PE

Response: We have revised as follows according to your advices:

(1) However, whether the d-dimer level is an ideal index to predict the prognosis of community-acquired pneumonia remains debatable, because no relevant studies focusing on d-dimer levels before treatment in patients with CAP specifically were available to conduct meta-analyses previously. For this reason, a meta-analysis was performed to systematically and quantitatively evaluate the prognostic accuracy of the d-dimer level before treatment in CAP. 

(2) All “data were” were replaced with “data was”

(3) “hospital dead cases” and “PE attack patients” were replaced with “non-survivors” and “patients with PE” 

(4) “to explore the distinguished role” was deleted.

(5) “PE high risk population” was replaced with high-risk population of PE

Background

Page 11

(1) Line 1: one of the most common respiratory diseases

(2) Line 4: CAP patients would progress into severe CAP

(3) Line 5: However, for a large proportion of CAP cases who visit emergency department could receive medical treatment without hospitalization.

(4) Line 7: critical for accurate treatment and to set setting up an appropriate care strategy

(5) To find reference 6

Line 12: procalcitonin [9, 10] had been reported with a great sensitivity for predicting the prognosis of CAP

(6) Line 14: Nevertheless, their specificities are not good enough to distinguish the various clinical outcomes of CAP. (reference?)

Response: We have revised as follows according to your advices:

(1) “one of the most common respiratory diseases” has been deleted.

(2) “CAP patients would progress into severe CAP” was replaced with “enter the severe stage”

(3) We revised this sentence as “A considerable proportion of patients with CAP in the emergency department can be treated as outpatients.”

(4) We have deleted this sentence.

(5) We have revised and only emphasized the potential prognostic value rather than sensitivity and specificity. 

(6) We have revised and only emphasized the potential prognostic value rather than sensitivity and specificity. 

Page 12

Line 21: The inclusion criteria were as followings: should be “as follows”.

Response: We have revised as your advice. Thanks very much.

Page 13

(1) Line 21: Besides, “The quality” of predictive and diagnostic study’s. “the quality”

Response: We have deleted this sentence.

Page 15

Line 12: The network will be evaluated using frequentist multivariate meta-analysis. Should be “the network was evaluated”

Response: We have replaced “will be” with “was”.

Page 16

(1) Line 4: after carefully evaluated or after careful evaluation.

(2) Line 10: In addition, patients with diarrhea and congenital heart diseases were excluded in 3 studies of children. (not clear)

Response: 

(1) We have revised as your advice.

(2) We have deleted this sentence.

Page 17

(1) Line 25: ORs had been pooled to identify.

Data analysis and interpretation was carried out properly.

Page 21 line 17

elevated D-dimer is an available supplement to PSI and CURB-65, which are the most widely used points-scoring system to assess severity of pneumonia. what is noted from reference 50 is "The coagulation system is often activated in CAP, and development of thrombocytopenia (platelet count, <100,000 cells/mm3) is also associated with a worse prognosis [86, 90–92]."-

Is it quite certain with this reference only to use "is an available supplement to PSI and CURB-65 with this reference only "?

Response: 

(1) We have revised as “Pooled results from adjust ORs have identified”.

(2) We agree with your opinion and have deleted this sentence because it is not convincing that d-dimer is an available supplement to PSI and CURB-65 with this reference.

---

## [Decision Letter · Decision Letter 1]

16 Jul 2021

PONE-D-20-17727R1

Value of D-dimer in predicting various clinical outcomes following community-acquired pneumonia: A network meta-analysis

PLOS ONE

Dear Dr. Li,

Thank you for submitting your manuscript to PLOS ONE. After careful consideration, we feel that it has merit but does not fully meet PLOS ONE’s publication criteria as it currently stands. Therefore, we invite you to submit a revised version of the manuscript that addresses the points raised during the review process.

We look forward to receiving your revised manuscript.

Kind regards,

Cho Naing, MBBS, PhD, FRCP

Academic Editor

PLOS ONE

Journal Requirements:

Additional Editor Comments (if provided):

The manuscript has been improved to a certain extent.

Still, there are important areas needed to improve.

As indicated by the Reviewer # 1, it is not meaningful to combine data from the different study designs.

Thank you.

Reviewers' comments:

Reviewer's Responses to Questions

**Comments to the Author**

1. If the authors have adequately addressed your comments raised in a previous round of review and you feel that this manuscript is now acceptable for publication, you may indicate that here to bypass the “Comments to the Author” section, enter your conflict of interest statement in the “Confidential to Editor” section, and submit your "Accept" recommendation.

Reviewer #1: (No Response)

Reviewer #2: All comments have been addressed

2. Is the manuscript technically sound, and do the data support the conclusions?

Reviewer #1: No

Reviewer #2: Yes

3. Has the statistical analysis been performed appropriately and rigorously? 

Reviewer #1: No

Reviewer #2: Yes

4. Have the authors made all data underlying the findings in their manuscript fully available?

Reviewer #1: Yes

Reviewer #2: Yes

5. Is the manuscript presented in an intelligible fashion and written in standard English?

Reviewer #1: Yes

Reviewer #2: Yes

6. Review Comments to the Author

Reviewer #1: I have seen a lot of improvements since the first draft of the manuscript. However, methodologically speaking, I still have major doubts.

1. The authors identified only 1 RCT and the rest are observational study. I don't think it is reasonable to mix results of RCT and observational studies into a meta-analysis. They should be reported, analyzed, discussed, and concluded separately because their data were generated from totally different study designs.

2. SMD is standardized mean difference not "standard mean deviation" as it was mentioned in the abbreviation list. This is used for measuring effect of treatment. I am really not sure if this is applicable for prognostic study. It is unclear how the concept of "mean difference" is applied to determine whether d-dimer can be used to predict outcome. I feel it is more straightforward to use OR for this purpose, where a standard cut-off is used to determine high/low d-dimer level.

3. Variations in methodologies used to measure d-dimer level might have contributed to the significant heterogeneity. The authors have mentioned of doing subgroup analyses. I wonder if variations in methodologies were considered when doing subgroup analyses?

Reviewer #2: 1) pg 17 why significant heterogeneity is mentioned as P<0.1, Did this value apply to the whole manuscript?

2) pg 19

Assessment of methodological quality

QUIPS tool had been used to assess the quality of included studies (Table 2). Among all the 32 studies, 20 were high risk of bias [11, 30, 31, 33, 34, 36-41, 43, 48, 53-55, 57-60] and only 2 studies were evaluated as low risk of bias[42, 52]. The number of individual domains rated as high risk ranged from 0 to 3.

Is there any specific reason that why moderate category is excluded from the discussing in the “Data Extraction and Study Quality Assessment component as well as from the “results” session.

3) Results session

- kindly recheck the following data:

o pooled result of figure 4 SMD=1.22 95%CI 0.67-1.77, I2=85.1% p=0.000 written in the paragraph and the data from the figure

o supplementary figure 4 – 3 studies or 4 studies, pooled result

"Pooled results of adjusted ORs from 4 studies [50-52, 54] have confirmed the prognostic value of increased D dimer (ES=0.90 95%CI 0.62-1.17, I2=59.4% p=0.008) (Supplementary Figure. 4). We didn’t find factors contributing to the heterogeneity by subgroup analysis"

Kindly recheck which was written in the paragraph and the data from the figure

o pooled result of supplementary figure 6

Pooled effect based on continuous data from three studies suggest a significant elevation in CAP patients with PE than those without PE (ES=1.07 95%CI 1.01-1.13, I2=60.2% p=0.113). (Supplementary Figure. 6).

kindly recheck which was written in the paragraph and the data from the figure

o some data of supplementary figure 7

I2=81.4%

kindly recheck which was written in the paragraph and the data from the figure

- for the “others” kindly refer with the respective figures

Pooled effects suggested patients with CAP had a higher level of d-dimer compared to healthy participants (SMD=0.88, 95%CI 0.54-1.22, I2=57.9 %, p=0.037) and D-dimer was significantly further elevated in patients requiring invasive mechanical ventilator (SMD=1.01, 95%CI 0.69-1.33, I2=0.0%, p=0.815).

(Kindly add "Referencing figures")

4) page 22

- 2nd paragraph – “this is may be” – incorrect use, please update

5) I would suggest to avoid using “it’s” “didn’t” – instead kindly use "it is or did not" and some minor punctuation and grammar usage.

7. PLOS authors have the option to publish the peer review history of their article (what does this mean?). If published, this will include your full peer review and any attached files.

Reviewer #1: No

Reviewer #2: No

---

## [Author Response · Author response to Decision Letter 1]

1 Oct 2021

Reviewer 1

1. The authors identified only 1 RCT and the rest are observational study. I don't think it is reasonable to mix results of RCT and observational studies into a meta-analysis. They should be reported, analyzed, discussed, and concluded separately because their data were generated from totally different study designs.

Response to reviewer:

We are sorry for this point of confuse. The reference 42 (Snijders, 2012), which was mentioned as one RCT in our article, was a secondary analysis based on a randomized controlled trial. Although the original analysis was in line with principles of RCTs, the secondary analysis based on outcome of patients (for example, d-dimer value of survivor and non-survivors) should be regarded as case-control study design. We have revised its design type as case-control study and examined the robustness of conclusion without this study which was not showed in article. Thanks very much.

2. SMD is standardized mean difference not "standard mean deviation" as it was mentioned in the abbreviation list. This is used for measuring effect of treatment. I am really not sure if this is applicable for prognostic study. It is unclear how the concept of "mean difference" is applied to determine whether d-dimer can be used to predict outcome. I feel it is more straightforward to use OR for this purpose, where a standard cut-off is used to determine high/low d-dimer level.

Response to reviewer:

It’s our negligence in abbreviation list where we wrongly used “standard mean deviation” and we have corrected it. As a guide for meta-analysis of prognostic studies published in BMJ journal suggested, unadjusted and adjusted prognostic effect estimates (eg, risk ratios, odds ratios, hazard ratios, mean differences) could be results to examine prognostic factors, although adjusted ratios are most recommended. The unadjusted odds ratio which we have transformed to standardized mean difference represented the risk that patients with poor outcome or prognosis had high levels of d-dimers. TAmong included studies for our meta-analysis, only a small proportion of them had supplied adjusted ORs and we have made data synthesis of these ORs. Because no consensus for prognostic meta-analysis, we also examined efficacy of prognostic factors with the help of principle of diagnostic meta-analysis, which regarded patients with poor outcome as positive evens of gold standard and regarded prognostic factors with cutoff as diagnostic method. Dose-response meta-analyses were a relatively ideal method to take full advantage of information and data of primary studies. Unfortunately, undetailed reports of included primary studies made it impossible to make dose-response curves. To be honest, our original intention was to measure the risk ratios that patients with different levels of d-dimer had poor outcome, but the existing studies did not support to complete such a meta-analysis. Therefore, we chose to transform odds ratio to SMD, not vice versa, to directly represent higher d-dimer level in patients with poor outcome. On the other hand, most of included studies reports continuous data and too much transform would bring unpredicted bias. We expected that our work would inspire more researchers to make more scientific studies and report more detailed data. We are trying to make a meta-analysis of individual participant data to solve this problem. Thank you. 

3. Variations in methodologies used to measure d-dimer level might have contributed to the significant heterogeneity. The authors have mentioned of doing subgroup analyses. I wonder if variations in methodologies were considered when doing subgroup analyses?

Response to reviewer:

We are sorry for absence of mention of this subgroup analysis. We had made subgroup meta-analysis by methodologies of d-dimer measurement, but we didn’t put the figure of synthesis into our article because of too many more important figures. When pooled results showed great heterogeneity, nearly all subgroups including more than 3 studies also have considerable heterogeneity. For example, I-square of subgroups of studies having used Immunoturbidimetric assay, latex immunoassay and ELISA were 86.1%, 96.4% and 96.1%, respectively. We have added this point in our article.

Reviewer #2:

 1) pg 17 why significant heterogeneity is mentioned as P<0.1, Did this value apply to the whole manuscript?

Response to reviewer:

A limitation of Cochran’s Q test is that it might be underpowered when few studies have been included or when event rates are low. Therefore, it is often recommended to adopt a higher P-value (rather than 0.05) as a threshold for statistical significance when using Cochran’s Q test to determine statistical heterogeneity. We added relevant reference for this P value in our article.

(2) pg 19

Assessment of methodological quality 

QUIPS tool had been used to assess the quality of included studies (Table 2). Among all the 32 studies, 20 were high risk of bias [11, 30, 31, 33, 34, 36-41, 43, 48, 53-55, 57-60] and only 2 studies were evaluated as low risk of bias [42, 52]. The number of individual domains rated as high risk ranged from 0 to 3.

Is there any specific reason that why moderate category is excluded from the discussing in the “Data Extraction and Study Quality Assessment component as well as from the “results” session.

Response to reviewer:

We are sorry for our confusing description. What we wanted to make readers knew was that 20 included studies were high risk, 2 were low risk and the remaining 10 studies were medium risk of bias. We have revised it. Thank you. 

3) Results session

- kindly recheck the following data:

o pooled result of figure 4 SMD=1.22 95%CI 0.67-1.77, I2=85.1% p=0.000 written in the paragraph and the data from the figure

o supplementary figure 4 – 3 studies or 4 studies, pooled result

"Pooled results of adjusted ORs from 4 studies [50-52, 54] have confirmed the prognostic value of increased D dimer (ES=0.90 95%CI 0.62-1.17, I2=59.4% p=0.008) (Supplementary Figure. 4). We didn’t find factors contributing to the heterogeneity by subgroup analysis"

Kindly recheck which was written in the paragraph and the data from the figure

o pooled result of supplementary figure 6

Pooled effect based on continuous data from three studies suggest a significant elevation in CAP patients with PE than those without PE (ES=1.07 95%CI 1.01-1.13, I2=60.2% p=0.113). (Supplementary Figure. 6).

kindly recheck which was written in the paragraph and the data from the figure

o some data of supplementary figure 7

I2=81.4%

kindly recheck which was written in the paragraph and the data from the figure

- for the “others” kindly refer with the respective figures

Pooled effects suggested patients with CAP had a higher level of d-dimer compared to healthy participants (SMD=0.88, 95%CI 0.54-1.22, I2=57.9 %, p=0.037) and D-dimer was significantly further elevated in patients requiring invasive mechanical ventilator (SMD=1.01, 95%CI 0.69-1.33, I2=0.0%, p=0.815).

(Kindly add "Referencing figures")

Response to reviewer:

We are sorry for our serious negligence to mistakenly correspond valued of pooled results to figures. Thanks for your correction with patience and we have revised our article according to your suggestion. Thank you very much.

4) page 22

- 2nd paragraph – “this is may be” – incorrect use, please update

Response to reviewer:

We have modified it to “this is partly because of …”. Thanks for your advice.

5) I would suggest to avoid using “it’s” “didn’t” – instead kindly use "it is or did not" and some minor punctuation and grammar usage.

Response to reviewer:

We have replaced with “it is or did not” in our revised article. Thank you.

---

## [Decision Letter · Decision Letter 2]

17 Jan 2022

Value of D-dimer in predicting various clinical outcomes following community-acquired pneumonia: A network meta-analysis

PONE-D-20-17727R2

Dear Dr. Li,

We’re pleased to inform you that your manuscript has been judged scientifically suitable for publication and will be formally accepted for publication once it meets all outstanding technical requirements.

Kind regards,

Cho Naing, MBBS, PhD, FRCP

Academic Editor

PLOS ONE

Additional Editor Comments (optional):

The authors have addressed the comments with reasonable improvement and justification,

Thank you

Reviewers' comments:

Reviewer's Responses to Questions

**Comments to the Author**

1. If the authors have adequately addressed your comments raised in a previous round of review and you feel that this manuscript is now acceptable for publication, you may indicate that here to bypass the “Comments to the Author” section, enter your conflict of interest statement in the “Confidential to Editor” section, and submit your "Accept" recommendation.

Reviewer #1: All comments have been addressed

Reviewer #2: All comments have been addressed

2. Is the manuscript technically sound, and do the data support the conclusions?

Reviewer #1: Partly

Reviewer #2: Yes

3. Has the statistical analysis been performed appropriately and rigorously? 

Reviewer #1: Yes

Reviewer #2: Yes

4. Have the authors made all data underlying the findings in their manuscript fully available?

Reviewer #1: Yes

Reviewer #2: Yes

5. Is the manuscript presented in an intelligible fashion and written in standard English?

Reviewer #1: No

Reviewer #2: Yes

6. Review Comments to the Author

Reviewer #1: I have no more substantial comments for this manuscript, but I still see quite a number references to RCT while the authors have mentioned that no RCTs were included.

Reviewer #2: I am grateful for this well written manuscript and robust analysis. The comments have been appropriately addressed and the contents were revised accordingly.

7. PLOS authors have the option to publish the peer review history of their article (what does this mean?). If published, this will include your full peer review and any attached files.

Reviewer #1: No

Reviewer #2: No

---

## [Editor Report · Acceptance letter]

28 Jan 2022

PONE-D-20-17727R2 

Value of D-dimer in predicting various clinical outcomes following community-acquired pneumonia: A network meta-analysis 

Dear Dr. Li:

I'm pleased to inform you that your manuscript has been deemed suitable for publication in PLOS ONE. Congratulations! Your manuscript is now with our production department. 

Kind regards, 

on behalf of

Professor Cho Naing 

Academic Editor

PLOS ONE